# Near-infrared transillumination imaging combined with aperture photometry for the quantification of melanin in the iris pigment epithelium

**Maciej Czepita** *

2nd Department of Ophthalmology, Pomeranian Medical University, Szczecin, Poland

* maciej@czepita.pl

**Data Availability Statement:** The software used in this study can be freely downloaded by using this link: https://figshare.com/projects/Aperture_

## Abstract

Near-infrared transillumination is used in the diagnosis and the management of different eye diseases. In particular, it enables the visualization of melanin in the pigment epithelium of the iris. This technique is valuable in such conditions as pigment dispersion syndrome and Adie's tonic pupil. Thus, objective quantification of the amount of melanin shedded from the iris pigment epithelium may help in the management of these conditions. By combining aperture photometry with near-infrared iris transillumination this can be achieved. A total of 4 patients (7 eyes) were examined. Three patients were diagnosed with pigment dispersion syndrome in both eyes. One patient had Adie's tonic pupil in one eye. Near-infrared iris transillumination was performed by using a prototype apparatus. Aperture photometry measurements were carried out through specially developed software. The signal-to-noise ratio of the prototype apparatus was 52 dB (399:1). Each pixel within the near-infrared transillumination image corresponded with an area size of the iris of 85 μm x 83 μm. Measurements were taken from several points of the iris in all patients. The average aperture photometry value of transillumination defects was 1321.53 (ADU) ± 501.08 SD, while the average aperture photometry value of the papillary ruff was 90.83 (ADU) ± 53.4. On average transillumination defects transmit 14.55 times more near-infrared light than the papillary ruff. A prototype apparatus for the capture of near-infrared iris transillumination images and custom software enabling aperture photometry measurements of the obtained images has been developed for the purpose of this study. This study demonstrates a potential application of this technique in the diagnosis and management of patients with such conditions as pigment dispersion syndrome and Adie's tonic pupil.

## Introduction

Transillumination is a medical diagnostic technique of tissue illumination by transmission of light through the examined tissue [1–6]. In ophthalmology transillumination is used in the diagnosis and management of several conditions. Most notably in the localization of

Photometry_Tool_For_Infrared_Iris_
Transillumination/74913.

**Funding:** The author received no specific funding
for this work.

**Competing interests:** The author has declared that
no competing interests exist.

intraocular tumors [7], ciliary body cysts [8] and the diagnosis of pigment dispersion syndrome (PDS) [9], pigmentary glaucoma and Adie's tonic pupil [10].

Aperture photometry is a basic image analysis technique of measuring the flux from a light source within a predefined region of interest referred to as an aperture [11]. A representation of the object's flux is calculated by summing up all the pixel values within the aperture after subtracting an estimate of the background flux from pixel values of a background annulus centered on the aperture.

The aim of this study is to describe near-infrared iris transillumination imaging combined with aperture photometry in the quantification of melanin within the iris pigment epithelium. Due to a lack of commercially available equipment for this purpose, a prototype near infrared iris transillumination system and special aperture photometry software were developed. Special attention has been placed on measuring the amount of melanin present within the iris pigment epithelium of patients with pigment dispersion syndrome and Adie's tonic pupil. In these conditions significant loses of melanin within the iris pigment epithelium occurs. In pigment dispersion syndrome it is currently believed that loss of melanin from the iris pigment epithelium occurs due to midperipheral posterior concavity of the iris [12]. During physiological pupillary movement, rubbing of the iris pigment epithelium against the anterior zonular bundles occurs in these patients. This leads to shedding of pigment, which gets transported with the movements of aqueous humor. The shedded pigment may deposit on the posterior capsule of the lens or on the corneal endothelium, anterior surface of the iris and also on the trabecular meshwork of the iridocorneal angle. The obstruction to the outflow of aqueous humor caused by shedded pigment on the trabecular meshwork of the iridocorneal angle leads to an increase in intraocular pressure, which in turn leads to the development of pigmentary glaucoma. Loss of the iris pigment appears clinically as midperipheral, radial, slitlike pattern of transillumination defects. These defects can be occasionally seen by retroillumination in a slit lamp or by using a transilluminator. However, near-infrared iris transillumination provides the most sensitive method of detection [13]. Quantifying the amount of melanin shedded from the iris in patients with pigment dispersion syndrome and pigmentary glaucoma has been previously done by measuring the sizes of transillumination defects from near-infrared photographic images and also by measuring the amount of melanin granules present in the aqueous humor by means of the cell count mode of the laser flare-cell meter [14,15,16]. However, no observations using aperture photometry of near-infrared iris transillumination images were carried out previously.

In the case of Adie's tonic pupil, melanin in the iris pigment epithelium layer is believed to be shedded as a result of segmental iris atrophy [10]. These areas can be only clinically detected through iris transillumination. They are located near the pupillary margin and usually have a circular shape. Near-infrared iris transillumination provides the most sensitive method of detection. Quantifying the amount of melanin shedded has only been done by measuring the sizes of near-infrared transillumination defects [17]. No observations using aperture photometry of near-infrared iris transillumination images have been done previously.

Near-infrared iris transillumination was first used in 1977 by Saari et al. [18] in patients with Fuchs heterochromatic iridocyclitis. Following this study, Alward et al. [9] described the use of this technique in 1990 in detecting posterior iris melanin loss in patients with pigment dispersion syndrome and pigmentary glaucoma. Further studies on the potential use of this method in other conditions were later carried out [19, 20]. Modification of the equipment have also been described. Due to the properties of melanin within the iris pigment epithelium, light in the near-infrared is absorbed less than in the visible making iris transillumination defects more prominently visible. Therefore, measuring the flux of the near-infrared light within iris transillumination defects through aperture photometry enables the quantification

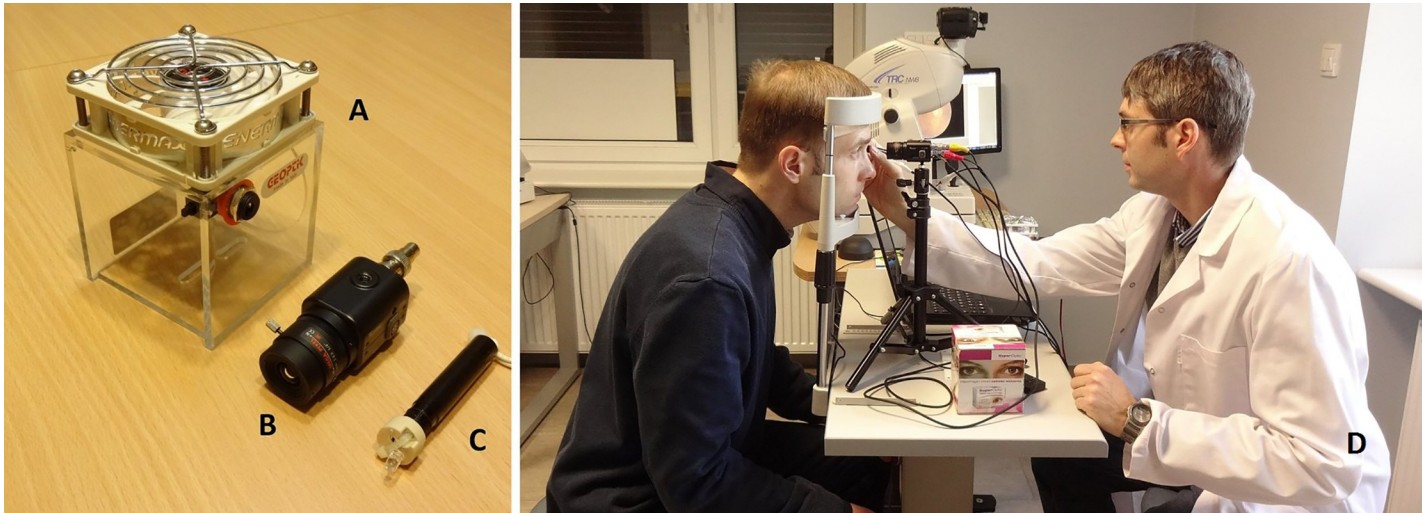

**Fig 1. Main components of prototype apparatus.** A–cooling fan. B–camera with lens. C–near-infrared transilluminator. D–near-infrared iris transillumination examination conducted by the author.

of the amount of melanin present in the iris pigment epithelium. The areas with less melanin have a higher flux value and vice versa.

## Methods

### Study participants

A total of 4 patients (7 eyes) were examined. Three of the patients had pigment dispersion syndrome, while one patient suffered from Adie's tonic pupil in one eye. Informed consent was obtained from each participant. The study followed the tenets of the Declaration of Helsinki and was approved by the Bioethics Committee of the Pomeranian Medical University. The individual pictured in Fig 1 has provided written informed consent (as outlined in PLOS consent form) to publish their image in this article.

### Description of prototype apparatus

The prototype near-infrared iris transillumination apparatus consists of a monochrome digital video camera sensitive to near-infrared light (WAT-910HX/RC, Watec Co., Ltd., Tsuruoka, Japan) with a macro lens (80CS20-50, 5mm, f2.2, LENEX). The camera with the lens is attached to a standard 1/4" camera tripod and placed in front of the patient on an ophthalmic table with a head and chin rest apparatus. The examination is conducted in a dark room. As a source of infrared illumination a modified MAGLITE Solitaire LED flashlight (Mag Instrument Inc., Ontario, CA, U.S.A.) is used (Fig 1). The modification involved removing the original light-emitting diode (LED) from the flashlight and replacing it with a 940 nm infrared LED diode. In order to lower the amount of thermal emission from the camera and thereby reduce the dark current of the camera a cooling fan is used (Geoptik, San Giovanni Lupatoto, Italy). The camera is placed inside the cooling fan for 10 minutes. The cooling fan is switched on. Afterwards the camera is removed from the cooling fan and connected to the tripod and objective. The patient is seated. The tip of the flashlight is gently pressed against the skin of the lower eyelid and the focus of the camera adjusted manually. A video recording of the procedure is made and stored on a laptop computer connected to the camera through a universal serial bus (USB) cable. The video is recorded for every examination with the corresponding

Patient-ID-Number with SharpCap software version 3.0 (AstroSharp Ltd., Harwell, U.K.). Video is edited through Free Video to JPG Converter (Digital Wave, Ltd., London, U.K.). An exemplary near-infrared iris transillumination image can be seen in Fig 2. Aperture photometry of the images is performed using the custom made Aperture Photometry Tool for Infrared Iris Transillumination Imaging software -APTITI (2nd Department of Ophthalmology, Pomeranian Medical University, Szczecin, Poland). The results of aperture photometry measurements are given in analog to units (ADU).

### Retinal irradiance calculation

To assess the potential hazard to the retina by the infrared transilluminator of the prototype apparatus the retinal irradiance was calculated.

Retinal irradiance was calculated according to the following equation [21]:

$$Er = \frac{\pi \, Ls \, \tau \, de^2}{4f^2}$$

$Er$ is the retinal irradiance (mW/cm$^2$), $Ls$ is the source irradiance (mW/cm$^2$ sr), $f$ is the effective focal length of the eye (in centimeters), $de$ is the pupil diameter (in centimeters) and $\tau$ is the transmittance of the ocular media. The source irradiance ($Ls$) of the infrared LED used was 60 mw/cm$^2$ sr. The transmittance $\tau$ of the ocular media is estimated to be 0.9 in normal patients[22]. A pupil diameter ($de$) of 0,7 cm was used as the examination was carried out under scotopic conditions in a dark room. The pupil dilates under these conditions to this size. The effective focal length of the eye $f = 1.7$ cm was used according to the model eye of

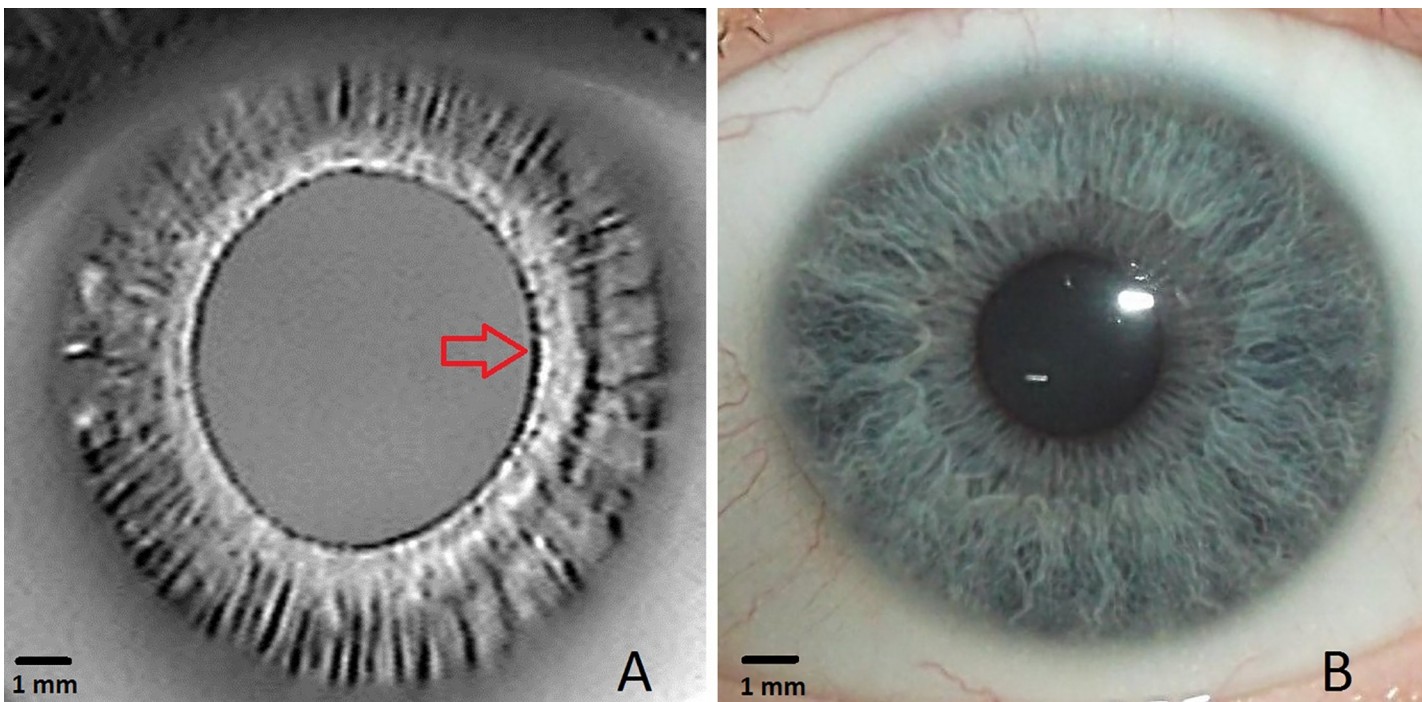

**Fig 2. Near-infrared iris transillumination image of the left eye of a healthy volunteer.** (A). The pupillary ruff can be seen as a narrow dark circular band at the pupillary margin (red arrow). For comparison a visible light image of the iris of the same eye can be seen (B).

Gullstrand [23]. Therefore $Er$ in this study was:

$$Er = 0.27 Ls\,\tau\,de^2 = 7.14\ mW/cm^2$$

According to the International Commission of Non-Ionizing Radiation Protection (ICNIRP) the retinal exposure limit to near-infrared light (IR-A radiation) is 6 W/(cm$^2$ sr) and the irradiance limit on the retina is 700 mW/cm$^2$ [24]. The calculated $Er$ of 7.14 mW/cm$^2$ is well below the threshold value of 700 mW/cm$^2$. Therefore, no damage to the retina with the proper use of the prototype apparatus should be expected.

## Compensating for defects within the CCD sensor of the camera

It is generally difficult to obtain a charge-coupled device (CCD) sensor free of image defects. Evaluating what kind of defects were present in our camera was done by cooling the camera for 10 minutes and afterwards obtaining a 10 second dark frame and a 10 second flat field image. Several hot pixels were noticed in the dark frame. These are pixels with higher than normal dark current. By superimposing the dark frame from the images obtained from our patients the exact positions of the hot pixels could be seen (Fig 3). Aperture photometry measurements using the Aperture Photometry Tool for Infrared Iris Transillumination Imaging were not performed at these locations in order not to obtain a false reading. The flat field image was recorded to spot any variabilities in sensitivities across the sensor. Variations in brightness were noticed at the edges of the flat field image due to vignetting. Dividing the near-infrared iris transillumination image pixel by pixel, by the obtained flat field image effectively removed these variations. However, through this process the connection of the pixel intensities to the collected photons became lost. Therefore, flat fielding was not applied to the obtained images. As a means to minimize the dark current of the CCD sensor of the camera we employed a thermo-electric cooler. The camera operating at room temperature was cooled by 9 degrees C.

## Precision of measurement

The signal-to-noise ratio of the Watec WAT 910HX/RC is 52 dB—399:1. This means that the signal is 399 times greater than the noise level or in other words the noise constitutes for 0.25% of the measurement. Therefore, the precision of the aperture photometry measurements is very high. If for example a baseline aperture photometry score is 1000 ADU, the amount of noise is equal to 2,5 ADU.

On average near-infrared iris transillumination images were captured from a distance of 4.7 cm. Given that the camera sensor is ½" and the size of the objective used is 5 mm, the horizontal width field of view of the captured images was 60 mm while the vertical height field of view of the captured images was 40 mm. This equals to a pixel size of the captured image to be 85 μm x 83 μm.

This implies that the aperture photometry measurements taken by the prototype apparatus have an accuracy of 399:1 for a minimal area size of the iris of 85 μm x 83 μm.

# Results

## Image processing

Once image acquisition is completed the video is exported to the Free Video to JPG Converter. The converter extracts all the frames of the video and saves them as JPEG images 720x480 pixels in size. This enables the examiner to pick out the best images for further evaluation with the Aperture Photometry Tool for Infrared Iris Transillumination Imaging-APTITI (Fig 4).

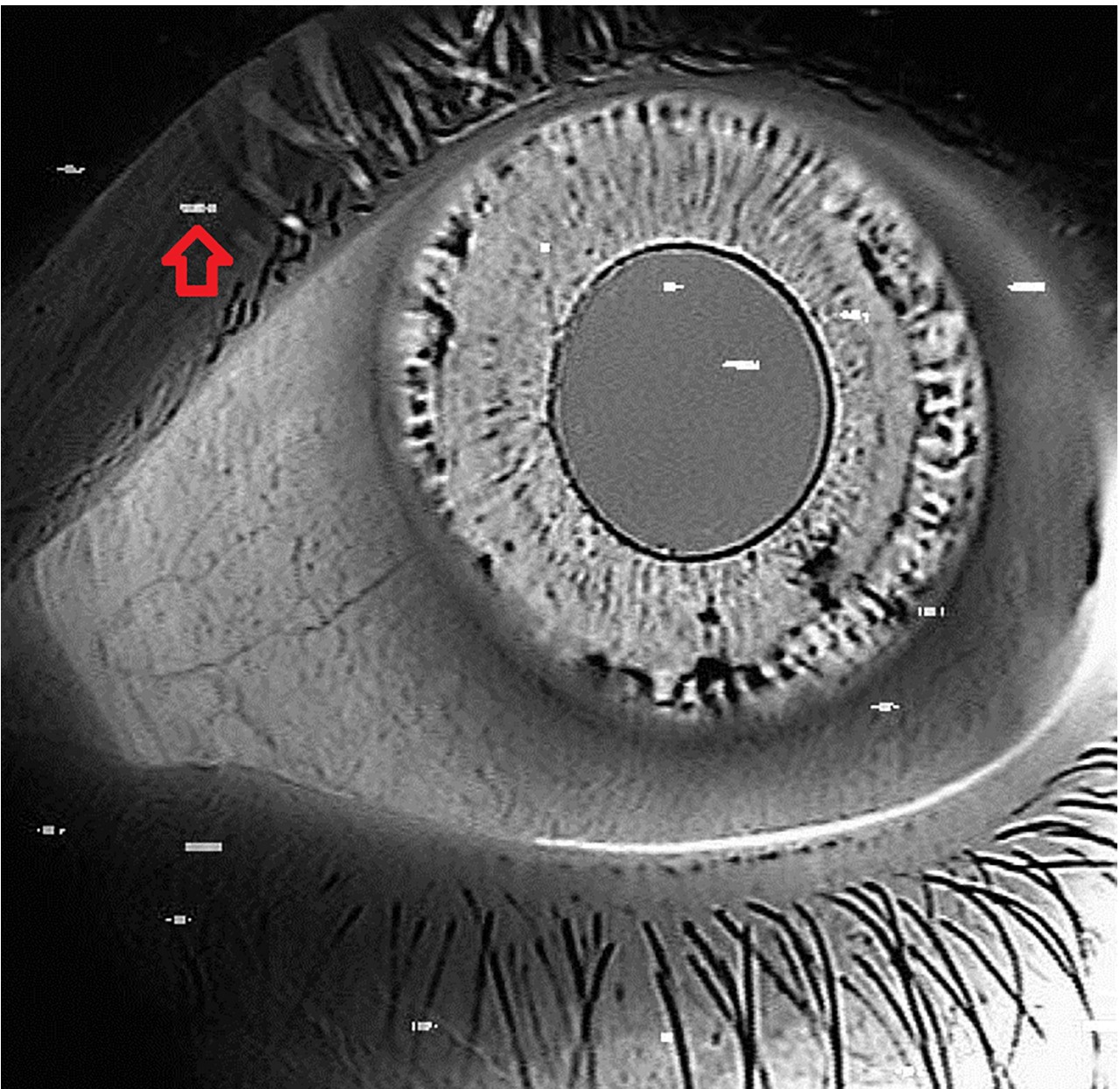

**Fig 3. Result of superimposing a dark frame image with a near-infrared iris transillumination image.** Many hot pixels can be seen (typical hot pixel marked by red arrow). No hot pixels were detected within the transillumination defects nor the pupillary ruff.

## Aperture photometry measurement with APTITI

The Aperture Photometry Tool for Infrared Iris Transillumination Imaging is a custom developed plugin written by the author of this study in JAVA for ImageJ (U.S. National Institutes of Health, Bethesda, Maryland, USA) image processing software. It is available for download along with an instruction manual at https://figshare.com/projects/Aperture_Photometry_Tool_For_Infrared_Iris_Transillumination/74913. Aperture photometry measurements of the near-infrared iris transillumination images are taken by clicking on the region of interest. A set of two apertures appear on the image–an inner and outer annulus. The size of these

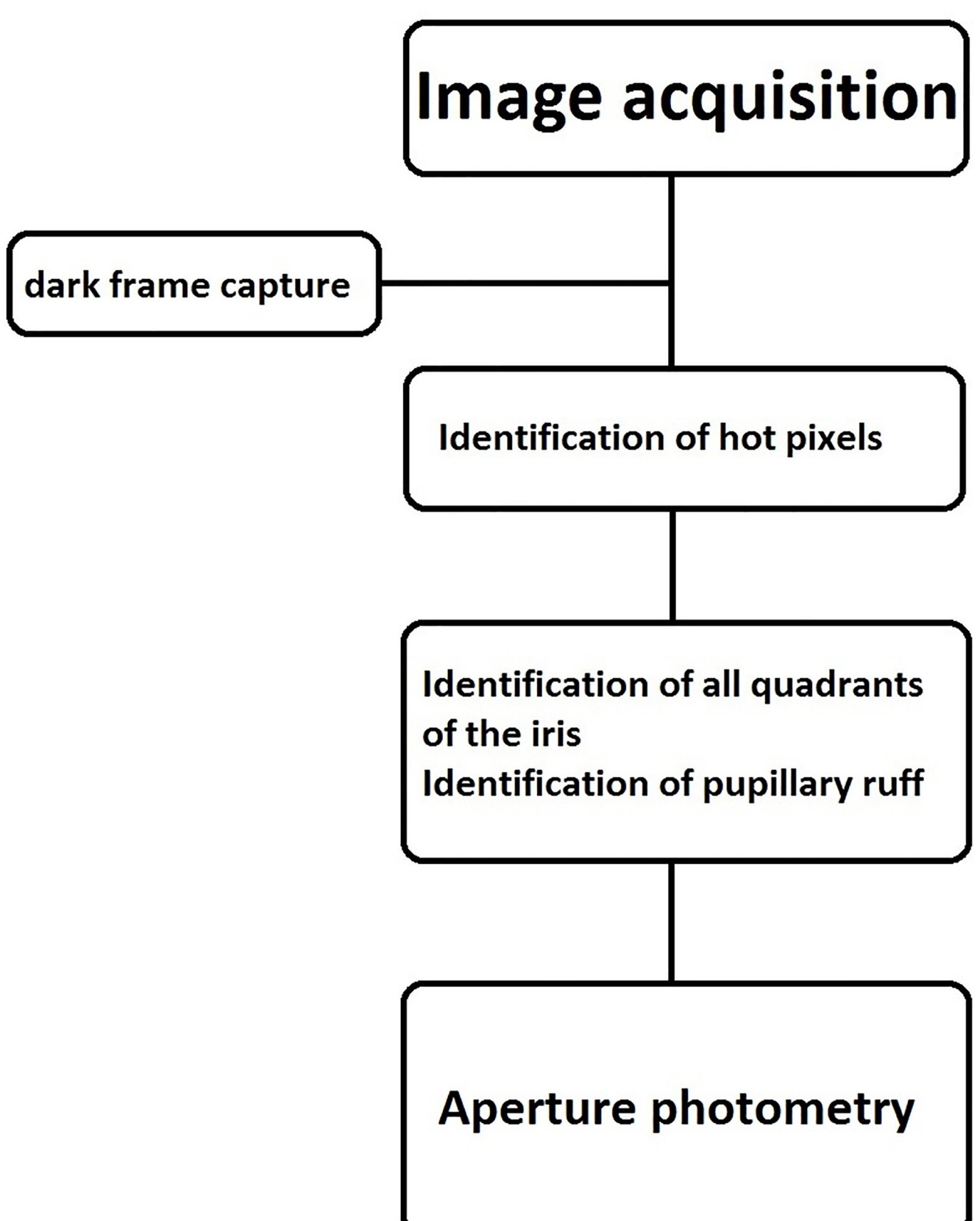

**Fig 4. Flowchart of image analysis algorithm.**

apertures can be adjusted. With both apertures of different sizes it is possible to get a net integrated count., which is the sum of all analog to digital units (ADU) after subtracting the background values from each pixel. In order to obtain the total counts without background subtraction, the radii of the inner and outer annulus need to be set to the same value. The value of the background will be zero. The obtained result will be the total of all the analog to digital units within the aperture. This second method was used in our measurements (S1 Fig).

In this study the iris is divided into four quadrants (Fig 5). The near-infrared iris transillumination defects are identified as well as the pupillary ruff. The pupillary ruff is a small portion of the iris pigment epithelium located at the edge of the pupil of the eye. It is not affected by the disease process in pigment dispersion syndrome and Adie's tonic pupil. It serves as a reference when performing aperture photometry. The measurements from the near-infrared iris transillumination defects are compared to the measurements from the pupillary ruff. This enables the quantification of the melanin present within the defects. An average value of the papillary ruff is calculated from measurements taken in different locations. This value is compared with the aperture photometry values of the transillumination defects. In Fig 5 (subject 1) the average aperture photometry value of the papillary ruff in the left eye is 61 ADU (Table 1). The average aperture photometry value from the upper right quadrant (A) is 2145 ADU. This implies that the transillumination defects in this quadrant transmit on average 35 times more infrared light than the pupillary ruff does. This signifies a significant loss of melanin within the iris pigment epithelium. The right eye in this patient remains not that severely affected. The average aperture photometry value from transillumination defects of all quadrants of the iris is 1226 ADU, which is only around 7,5 times higher than the papillary ruff. This difference between both eyes is most probably caused by the fact that this patient underwent a pars plana vitrectomy with silicone oil injection for a retinal detachment of the left eye. The examination was carried out 1 month after the operation. In the remaining patients such significant differences were not observed. These patients had never undergone any intraocular surgery. Overall, the average aperture photometry value of transillumination defects was 1321.53 (ADU) ± 501.08 SD, while the average aperture photometry value of the papillary ruff was 90.83 (ADU) ± 53.4. Transillumination defects transmit on average 14.55 times more near-infrared light then the papillary ruff. This information enables more precise identification of quadrants potentially at risk of greater loss of melanin form the iris pigment epithelium.

## Discussion

The design and operation of a prototype device for near-infrared transillumination imaging of the eye has been presented along with a technique for the quantification of the amount of melanin within the iris pigment epithelium. This new technique based on aperture photometry enables more precise measurement of the amount of melanin within the iris pigment epithelium as compared to techniques used in previous studies. Instead of simply measuring the size of transillumination defects in near-infrared images of the iris as done by Haynes et al. [16] it is now possible measure the amount of melanin within each defect by using aperture photometry. This is an advantage as surely iris transillumination defects must differ from each other by the amount of melanin within.

In this study it was observed that a very low intensity infrared light-emitting diode (LED) of only 60mW/cm$^2$ sr combined with the WATEC WAT-910HX/RC camera was sufficient to obtain good quality near-infrared transillumination images of the eye. The quantum efficiency of the camera sensor at the wavelength used (940 nm) was equal to 15%. This is almost twice the quantum efficiency level at this wavelength for conventional charge-couple device (CCD) sensors. The frame rate of the during video acquisition was set to 30 fps. This enables to

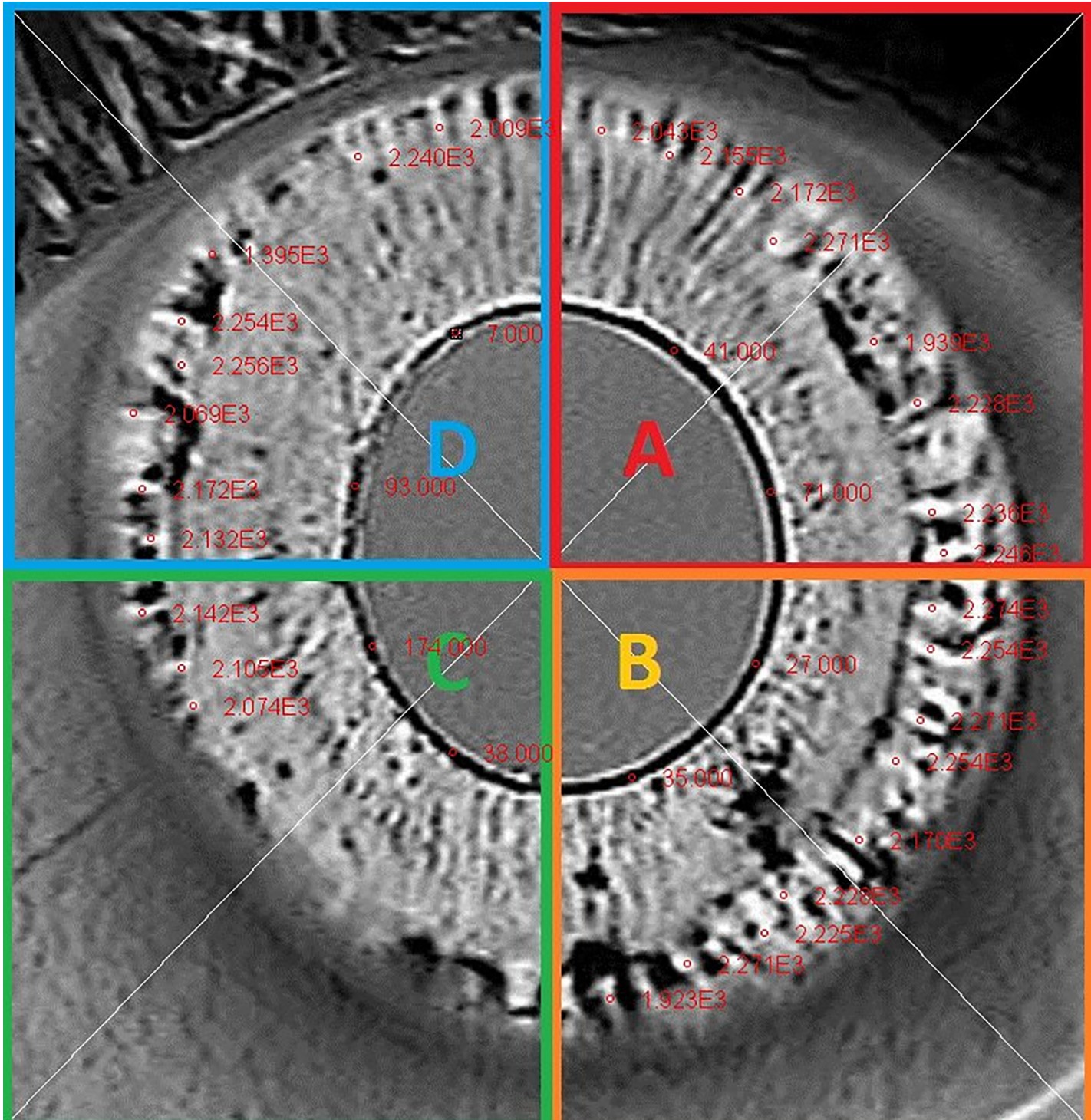

**Fig 5. Near infrared iris transillumination image of a patient with pigment dispersion syndrome (subject 1).** The iris is divided into four quadrants labeled as A,B,C and D respectively. Numerous bright spoke like iris transillumination defects are seen in the midperiphery of the iris in each quadrant. Aperture photometry measurements within the iris transillumination image can be seen (red circles) as well as their value.

**Table 1. Characteristics and results of patients examined in this study.**

| Subject Disease | Right eye | | | | | Left eye | | | | |
|---|---|---|---|---|---|---|---|---|---|---|
| Sex and age | Upper right quadrant average value in ADU | Lower right quadrant average value in ADU | Lower left quadrant average value in ADU | Upper left quadrant average value in ADU | Pupillary ruff average value in ADU | Upper right quadrant average value in ADU | Lower right quadrant average value in ADU | Upper left quadrant average value in ADU | Lower left quadrant average value in ADU | Pupillary ruff average value in ADU |
| Subject 1 PDS, M, 32 y.o. | 1117 | 1042 | 1298 | 1448 | 164 | 2145 | 2192 | 2065 | 2107 | 61 |
| Subject 2 PDS, M, 24 y.o. | 1534 | no defects | 1247 | no defects | 136 | 1226 | 1427 | 1421 | 997 | 103 |
| Subject 3 PDS, M, 40 y.o. | no data | no data | no data | no data | no data | no defects | 885 | no defects | 629 | 21 |
| Subject 4 Adie's, F, 33 y.o. | no defects | no defects | no defects | no defects | no data | 760 | 664 | 905 | no defects | 60 |

PDS–pigment dispersion syndrome, ADU–analog to digital units, M–male, F—female

capture on average around 100 frames in between each blinking of the eye. As a result of the high camera sensitivity and low intensity infrared light source used the retinal irradiance was also found to be insignificant with levels reaching only 1% of the maximal allowed norm according to the International Commission of Non-Ionizing Radiation Protection (ICNIRP). However, certain disadvantages of the camera also became apparent in this study. The main problem were defective pixels. Some hot pixels were discovered in the dark frame images. In order not to compromise the aperture photometry measurements these have to be avoided during video acquisition. This required lengthy preparation of the camera and patient positioning. A lesser problem was the signal-to-noise ratio of the camera. At 52 dB the signal-to-noise ratio (SNR) is somewhat high. However, given the high sensitivity of the camera (0.000005 lux) this was acceptable.

## Conclusions

A prototype apparatus for the capture of near-infrared transillumination images of the eye has been developed along with special JAVA based software for aperture photometry of the acquired images. This is the first time aperture photometry has been used in the quantification of melanin within the iris pigment epithelium. The design, operation and potential application of this new equipment have been described. The described method offering objective and reproducible quantification of melanin will allow for more accurate evaluation of the natural course and the effects of various treatment options in patients with different ocular conditions.

## Supporting information

**S1 Fig. Aperture photometry measurement of near-infrared iris transillumination defects using the Aperture Photometry Tool for Infrared Iris Transillumination Imaging.**
(TIF)

## Author Contributions

**Conceptualization:** Maciej Czepita.

**Formal analysis:** Maciej Czepita.

**Investigation:** Maciej Czepita.

**Software:** Maciej Czepita.

**Writing – original draft:** Maciej Czepita.

**Writing – review & editing:** Maciej Czepita.

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
