## [Decision Letter · Decision Letter 0]

8 Jan 2020

PONE-D-19-27146

Near-infrared transillumination imaging combined with aperture photometry for the quantification of melanin in the iris pigment epithelium

PLOS ONE

Dear Dr. Czepita,

Thank you for submitting your manuscript to PLOS ONE. After careful consideration, we feel that it has merit but does not fully meet PLOS ONE’s publication criteria as it currently stands. Therefore, we invite you to submit a revised version of the manuscript that addresses the points raised during the review process.

We would appreciate receiving your revised manuscript by Feb 22 2020 11:59PM. To enhance the reproducibility of your results, we recommend that if applicable you deposit your laboratory protocols in protocols.io, where a protocol can be assigned its own identifier (DOI) such that it can be cited independently in the future. For instructions see: http://journals.plos.org/plosone/s/submission-guidelines#loc-laboratory-protocols

We look forward to receiving your revised manuscript.

Kind regards,

Ireneusz Grulkowski, PhD

Academic Editor

PLOS ONE

Journal Requirements:

Reviewers' comments:

Reviewer's Responses to Questions

**Comments to the Author**

1. Is the manuscript technically sound, and do the data support the conclusions?

Reviewer #1: Partly

Reviewer #2: Partly

2. Has the statistical analysis been performed appropriately and rigorously? 

Reviewer #1: No

Reviewer #2: Yes

3. Have the authors made all data underlying the findings in their manuscript fully available?

Reviewer #1: Yes

Reviewer #2: Yes

4. Is the manuscript presented in an intelligible fashion and written in standard English?

Reviewer #1: Yes

Reviewer #2: Yes

5. Review Comments to the Author

Reviewer #1: Dear Maciej Czepita,

Thank you very much for choosing PLOS ONE journal to submit your research about „Near-infrared transillumination imaging combined with aperture photometry for the quantification of melanin in the iris pigment epithelium „. It is a useful paper since a clinical point of view. Nevertheless, I would like to ask you questions and give you some comments in order to make clear your research:

-In the introduction, there is short description about role of the melanin in Adie’s tonic pupil (lines 85-88). The appropriate reference about it is missing.

-„A total of 4 patiens (7 eyes) were examined. A total of 4 patients (7 eyes) were examined. Three of the patients had pigment dispersion syndrome, while one patient suffered from Adies’ tonic pupil in one eye. Informed consent was obtained from each participant „ (lines 108-109). More patients should be examined. Information about age and sex is missing. The healthy patients also should be examined (as a control).

- „Description of the prototype apparatus” – there should be scheme how the whole system with sample looks like during measurement – not only single parts of the set up (Fig.1).

- How much time takes examination of the one eye? If there is some kind of adaptation to dark conditions before measurement?

- Line 134: „An exemplary near-infrared iris transillumination image can be seen in Fig 2.” Should be indicate A or B.

-Fig.2. – The scale is missing. Better is to superimpose both images.

-„The transmittance τ of the ocular media is estimated to be 0.9 in normal patients” (lines 153-154) – how did you estimate the transmittance?

- „A pupil diameter (de) of 0,7 cm was used as the examination was carried out under scotopic conditions in a dark room.”(lines 154-155) – why exactly 0,7 cm?

-„ The effective focal length of the eye f = 1.7 cm was used according to the model eye of Gulstrand.” (lines 156-157) – the reference is missing

- „By superimposing the dark frame from the images obtained from our patients the exact positions of the hot pixels could be seen (Fig 3).” (lines 173-175) There should be arrows to indicate examples of hot pixels.

- „precision of measurements”, line 195, please explain in the text, what ADU means.

- Table – 1, please explain in the text, what PDS means.

Reviewer #2: The following old and recent papers on transillumination for different applications should be considered in the introduction:

DOI: 10.1109/10.817628

DOI: 10.1364/OL.15.001179

DOI: 10.1109/TBME.2003.812188

DOI: 10.3390/s19040851

DOI: 10.1016/j.biosystemseng.2014.06.014

DOI: 10.1016/j.compag.2019.02.014

Usually transillumination refers to a setup in which light source and detector are one in front of the other. In this case light source and detector are at 90°. Would you be able to comment on the differences? Are you actually detecting side scattered photons? Or is there another way to describe the phenomenon?

Why CCD and not a CMOS? There are CMOS sensors with enhanced responsitivy in the NIR.

Did you need dark room to ensure that the pupil is dilated or to avoid camera saturation with ambient light? In this second case, a narrow optical filter in front of the camera should be sufficient to select only the 940 nm photons

You need to define the acronyom ADU (Analog to Digital Unit) and explain the meaning of it in your experiments.

In many places the authors claim that their measurements enable the quantification of the

melanin. But I do not see any results reporting the exact amount of melanin. Only ADU values are reported but they are not indicating the amount of melanin. So the authors should solve this issue.

6. PLOS authors have the option to publish the peer review history of their article (what does this mean?). If published, this will include your full peer review and any attached files.

Reviewer #1: No

Reviewer #2: No

---

## [Author Response · Author response to Decision Letter 0]

29 Jan 2020

Response to remarks of Reviewer #1

1. I have now included an appropriate citation regarding the role of melanin in Adie’s tonic pupil (line 85-88).

2. I have now included information about the patients age and sex in Table 1. The overall incidence of pigment dispersion syndrome is thought to be around 4.8 per 100,000 population/year1. The overall incidence of Adie’s tonic pupil is approximately 4.7 per 100,000 population/year2. Both conditions are rare. The 4 patients participating in our study were recruited over a 2 year period. One patient newly diagnosed with pigment dispersion syndrome did not wish to participate in the study. A future study on a larger group of patients will be undertaken. The main aim of this paper is to demonstrate the feasibility of this technique in these conditions. A control group was not examined in this study as the transillumination defects are not present in healthy individuals. Therefore, there wouldn’t be anything to compare among the two groups.

3. I have now included an additional photograph in Figure 1 displaying the setup of the experimental apparatus during examination.

4. The examination takes around 1 minute per patient. Dark room adaptation before the examination was 1 minute.

5. I have now added indicated the A and B subpart in Figure 2.

6. I have added a scale in both subparts of Figure 2. It is not possible to superimpose both images of the iris in the visible and near-infrared because the pupil is larger in the near-infrared than in the visible. This is due to pupil dilatation in dark conditions and pupil constriction in light.

7. I have included now a citation on regarding the transmittance of the ocular media (line 153-154)

8. In the equation (line 149) we assumed a pupil diameter of 7 mm as this is the mean value for the examined age group.

9. The citation for the effective focal length of the schematic eye of Gullstrand has now been included (lines 156-157)

10. I have now added a red arrow in Figure 3 pointing to a hot pixel. I have added a explanation on ADU in the description of the prototype apparatus section of the manuscript.

11. I have now added an explanation of PDS in the manuscript.

References:

1.Ritch R, Steinberger D, Liebmann JM. Prevalence of pigment dispersion syndrome in a population undergoing glaucoma screening. Am J Ophthalmol. 1993 Jun 15;115(6):707-710.

2. Sarao MS, Sandeep S. Adie Syndrome https://www.ncbi.nlm.nih.gov/books/NBK531471/

Response to remarks of Reviewer #2

1. I have added in the introduction citations about the various medical uses of NIR transillumination.

2. The near-infrared transilluminator is held at an angle of around 60 degrees to the camera during the examination in order for the light to penetrate through the sclera and vitreous chamber and then reflect of the retina. The reflected light then causes backlighting of the iris. Therefore, the iris pigment epithelium can be imaged through this technique.

3. The WATEC WAT 910-HX/RC CCD camera was chosen because of it’s enhanced sensitivity in the NIR as well as the price which was within our budget for the study.

4. The examinations were carried out in a dark room in order avoid camera saturation with ambient light

5. The acronym ADU is now explained at first in the description of the prototype apparatus section of the manuscript.

6. In the study I compared the aperture photometry readings of the transillumination defects with the aperture photometry readings of the papillary ruff. The papillary ruff is the portion of iris pigment epithelium at the margin of the pupil. This part of the iris pigment epithelium is not affected by the disease process in pigment dispersion syndrome and usually also in Adie’s tonic pupil. By comparing the both results the amount of melanin shedded it can be estimated.

---

## [Decision Letter · Decision Letter 1]

25 Feb 2020

Near-infrared transillumination imaging combined with aperture photometry for the quantification of melanin in the iris pigment epithelium

PONE-D-19-27146R1

Dear Dr. Czepita,

We are pleased to inform you that your manuscript has been judged scientifically suitable for publication and will be formally accepted for publication once it complies with all outstanding technical requirements.

With kind regards,

Ireneusz Grulkowski, PhD

Academic Editor

PLOS ONE

Additional Editor Comments (optional):

Reviewers' comments:

Reviewer's Responses to Questions

**Comments to the Author**

1. If the authors have adequately addressed your comments raised in a previous round of review and you feel that this manuscript is now acceptable for publication, you may indicate that here to bypass the “Comments to the Author” section, enter your conflict of interest statement in the “Confidential to Editor” section, and submit your "Accept" recommendation.

Reviewer #1: All comments have been addressed

Reviewer #2: All comments have been addressed

2. Is the manuscript technically sound, and do the data support the conclusions?

Reviewer #1: (No Response)

Reviewer #2: (No Response)

3. Has the statistical analysis been performed appropriately and rigorously? 

Reviewer #1: (No Response)

Reviewer #2: (No Response)

4. Have the authors made all data underlying the findings in their manuscript fully available?

Reviewer #1: (No Response)

Reviewer #2: (No Response)

5. Is the manuscript presented in an intelligible fashion and written in standard English?

Reviewer #1: (No Response)

Reviewer #2: (No Response)

6. Review Comments to the Author

Reviewer #1: (No Response)

Reviewer #2: (No Response)

7. PLOS authors have the option to publish the peer review history of their article (what does this mean?). If published, this will include your full peer review and any attached files.

Reviewer #1: No

Reviewer #2: No

---

## [Editor Report · Acceptance letter]

27 Feb 2020

PONE-D-19-27146R1 

Near-infrared transillumination imaging combined with aperture photometry for the quantification of melanin in the iris pigment epithelium 

Dear Dr. Czepita:

I am pleased to inform you that your manuscript has been deemed suitable for publication in PLOS ONE. Congratulations! Your manuscript is now with our production department. 

With kind regards,

on behalf of

Dr. Ireneusz Grulkowski 

Academic Editor

PLOS ONE